# The High-Precision Liquid Chromatography with Electrochemical Detection (HPLC-ECD) for Monoamines Neurotransmitters and Their Metabolites: A Review

**DOI:** 10.3390/molecules29020496

**Published:** 2024-01-19

**Authors:** Bruno P. Guiard, Guillaume Gotti

**Affiliations:** 1Centre de Recherches sur la Cognition Animale (CRCA), CNRS UMR5169, 31062 Toulouse, France; bruno.guiard@univ-tlse3.fr; 2Centre de Biologie Intégrative (CBI), Faculté Sciences Ingénierie (FSI), Université de Toulouse III, 31062 Toulouse, France

**Keywords:** HPLC, electrochemical detection, monoamine, neurotransmitters, serotoninergic system, dopaminergic system

## Abstract

This review highlights the advantages of high-precision liquid chromatography with an electrochemical detector (HPLC-ECD) in detecting and quantifying biological samples obtained through intracerebral microdialysis, specifically the serotonergic and dopaminergic systems: Serotonin (5-HT), 5-hydroxyindolacetic acid (5-HIAA), 3,4-dihydroxyphenylacetic acid (DOPAC), dopamine (DA), 3-metoxytryptamin (3-MT) and homovanillic acid (HVA). Recognized for its speed and selectivity, HPLC enables direct analysis of intracerebral microdialysis samples without complex derivatization. Various chromatographic methods, including reverse phase (RP), are explored for neurotransmitters (NTs) and metabolites separation. Electrochemical detector (ECD), particularly with glassy carbon (GC) electrodes, is emphasized for its simplicity and sensitivity, aimed at enhancing reproducibility through optimization strategies such as modified electrode materials. This paper underscores the determination of limits of detection (LOD) and quantification (LOQ) and the linear range (L.R.) showcasing the potential for real-time monitoring of compounds concentrations. A non-exhaustive compilation of literature values for LOD, LOQ, and L.R. from recent publications is included.

## 1. Introduction

Monoamine neurotransmitters (NTs) are a class of NTs that include brain chemicals involved in the transmission of signals between nerve cells, or neurons. The two main types of monoamine NTs are catecholamines and indolamines: catecholamines include dopamine (DA), norepinephrine (NE), and epinephrine (E) and indolamines include serotonin (5-hydroxytryptamine or 5-HT).

These monoamine NTs play a crucial role in the regulation of various physiological and behavioral processes, such as the regulation of mood, sleep, appetite, reward, attention, and stress. Imbalances in the functioning of monoamine NTs are associated with various neurological and psychiatric disorders.

### 1.1. The Serotonergic System

The serotonergic system utilizes 5-HT as a neurotransmitter and plays a pivotal role in regulating diverse physiological and psychological processes, including mood, sleep, appetite, thermoregulation, and heart rate [1].

5-HT is synthesized from the essential amino acid tryptophan (TRP). Within neurons, TRP undergoes conversion to 5-hydroxytryptophan (5-HTP), catalyzed by the tryptophan hydroxylase enzyme. Subsequently, an enzymatic reaction involving aromatic amino acid decarboxylase (AADC) transforms 5-HTP into 5-HT (Figure 1). Following synthesis, the 5-HT is stored in vesicles via the vesicular monoamine transporter (VMAT) within the nerve endings of serotonergic neurons. Upon the arrival of an action potential at the nerve ending and the membrane depolarization, it triggers the release of 5-HT from vesicles into the synaptic cleft. Here, it can interact with receptors located on the postsynaptic membrane, known as serotonin receptors or 5-HT receptors. Extracellular 5-HT may also bind and activate inhibitory receptors expressed on 5-HT neurons (5-HT1A/1B/1D) thereby exerting negative feedback on its own release. Intracellular excess of 5-HT can be sequentially broken down by monoamine oxidase (MAO) and aldehyde dehydrogenase, resulting in the accumulation of 5-hydroxyindoleacetic acid (5-HIAA) in the intracellular space. All these receptors are present in the central nervous system and various tissues throughout the body. The effects of 5-HT vary based on the specific receptors it binds to.

Dysfunctions in the serotonergic system may contribute to several disorders, including neurodegenerative disease like Parkinson’s [2] or Alzheimer’s [3,4], and also psychiatric disorders such as anxiety [5] and depression [6].

### 1.2. The Dopaminergic System

The dopaminergic system is a neurotransmission system in the brain that uses DA as a neurotransmitter. DA is a chemical compound that plays a crucial role in various brain functions, including the regulation of motivation [7], pleasure, reward, and movement [8].

The dopaminergic system is composed of several neural pathways, with the most well-known being the mesolimbic, mesocortical, and nigrostriatal pathways. It plays a key role in various physiological and behavioral processes, and its dysfunction is associated with several neurological and psychiatric disorders. For example, overactivity of the dopaminergic system is linked to schizophrenia [9,10], while deficiency is associated with Parkinson’s disease [11]. Medications that modulate dopaminergic activity are often used to treat these disorders.

DA is synthesized in the neuron from tyrosine (TYR) through a two-step enzymatic reaction involving first, the tyrosine hydroxylase, synthesizing the levodopa (L-DOPA) and then, the DOPA decarboxylase. The DA can then be introduced into vesicles by the VMAT and released into the synaptic cleft. Degradation of DA can be catalyzed by MAO into DOPAC within the neuron. Two pathways are possible for its transformation into homovanillic acid (HVA), either through catecholamine-O-methyl-transferase (COMT) and MAO sequentially, with 3-metoxytryptamine (3-MT) as an intermediate, or by directly degrading DOPAC via COMT (Figure 2).

Two additional catecholamines, derived from dopamine and integral to the neurotransmitter family, are norepinephrine (NE) and epinephrine (E). E plays a dual role, functioning as a neurotransmitter within the central nervous system and as a hormone within the adrenal medulla. NE, conversely, acts as a metabolic precursor to epinephrine and operates as a neurotransmitter in the central nervous system. Both NE and E are crucial in regulating fundamental processes such as the sleep-wake cycle, vigilance, stress response [12], emotions [13], learning [14], and memory [15]. Their multifaceted roles underscore their significance in orchestrating key physiological and cognitive functions within the intricate framework of the human body.

## 2. Microdialysis Experiment

Microdialysis, employed for the past forty years [16,17], is a real-time sampling technique facilitating the analysis of extracellular NTs [18,19]. It functions on the principle of passive diffusion of small molecular-weight compounds through a porous membrane, transitioning from the region with the highest NTs concentration, namely the synaptic extracellular space, to the less concentrated area, the dialysis probe perfused with a neurotransmitter-free buffer solution at physiological pH. This technique allows the collection of dialysates at variable intervals, with flow rates ranging from 0.5 to 1.5 μL min^−1^, dependent on the experimental setup and brain area studied (Figure 3). These samples, which include monoamines and their metabolites, play a crucial role in comprehending the functioning of the serotonin and dopamine systems and associated disorders.

The reported sensitivity limit for 5-HT in various scientific publications is approximately 0.5 fmol per sample with a signal-to-noise ratio = 2 [20]. To comprehensively grasp the serotonergic and dopaminergic systems functionalities and its associated disorders and diseases, a combination of microdialysis sampling methods and high-performance analytical techniques is crucial for quantifying variations in extracellular NTs and metabolite concentrations.

## 3. Analytical Methods for Detection and Quantification of NTs and Metabolites

The analysis methods have been developed for the detection and quantification of NTs in vivo to understand specific neurotransmission dysfunctions in neurodegenerative diseases or in psychiatric disorders.

### 3.1. Gas Chromatography and Capillary Electrophoresis

Gas chromatography (GC) [21,22] and capillary electrophoresis (CE) [23,24] are analytical techniques used in these research areas. However, they have several drawbacks: GC requires the use of compound derivatization to make them more volatile [25], and CE is a limited technique for the analysis of complex mixtures. HPLC, on the other hand, allows for the analysis of biological samples from microdialysis without the need for chemical derivatization or extraction steps.

### 3.2. HPLC

Since the late 1980s, research teams led by Donzanti et al. [26], Peinado et al. [27], and Rogers et al. [28] have focused their studies on the detection of amino acid neurotransmitters, particularly γ-aminobutyric acid (GABA), achieving a sensitivity of about 0.1 nM. HPLC analysis of NTs is characterized by its speed and selectivity. Typically, a reverse-phase (RP) system is used to separate NTs due to their hydrophobic nature. For instance, Bidel et al. developed an HPLC method for the separation of monoamines and their metabolites (Figure 4a). They achieved a total elution time of about 12 min, a minimum resolution between each peak of 2.2, and a number of theoretical plates (N) ranging from 2900 to 3800 [29]. These values are largely sufficient to validate the use of HPLC.

By way of comparison, Reinhoud et al. developed a UHPLC method similar conditions (see Section 3.3 for more details) under reverse phase conditions, enabling the separation of NTs in a 12 min run time elution (Figure 4b). The columns of UHPLC have smaller particle sizes. Consequently, chromatographic peaks will be finer and better defined, allowing for improved separation of compounds. Moreover, the number of theoretical plates in this study has been estimated to be close to 200,000 [30].

More recently, hydrophilic interaction liquid chromatography (HILIC) columns have been employed [31,32]. These columns offer reduced system pressure, allowing for enhanced flow rates and shorter analysis times. For example, in the study by Zhou et al., the use of a low-granularity column (1.7 µm) combined with a flow rate of 0.4 mL min^−1^ is employed. These experimental conditions result in very short analysis times, on the order of 4 min. It is noteworthy that the Multiple Reaction Monitoring (MRM) mode of the mass spectrometer enhances the selectivity of the measurements [33].

The utilization of HILIC column necessitates coupling with mass spectrometry because of its incompatibility with the salts used with the electrochemical detection.

### 3.3. Detection of NTs and Their Metabolites

Various types of detectors can be employed, including fluorescence detectors (FLD) [34,35,36], mass spectrometry (MS) [37,38,39], sensors and biosensors [40,41,42] or electrochemical detector (ECD). ECD holds an advantage over MS detectors due to its lower operating cost and easy maintenance. For instance, weekly polishing of a glassy carbon (GC) working electrode with alumina powder is adequate to maintaining a reflective surface, thereby enhancing result repeatability and reproducibility.

ECD has also been extensively utilized for the detection and quantification of NTs and their metabolites [43,44,45,46]. It proves to be a straightforward and sensitive method for quantifying small amounts of NTs. One of its notable benefits is its simplicity and minimal maintenance.

Several types of electrochemical detectors exist, including coulometric and amperometric detectors. The principle of the coulometric detector lies in the measurement of the current charge produced during electrolysis at a constant potential. Compounds are then oxidized or reduced almost entirely. Coulometric detectors are less sensitive to variations in flow rate and temperature than amperometric detectors. However, electrodes are easily contaminated due to their large electrolysis surface area. The measurement of the charge during the electrochemical reaction is based on Faraday’s law (Equation (1)).
(1)$$Q=n\;F\;N_{O}$$
with Q the charge (C or A s), n the number of electrons exchanges during the electrochemical detection, F the Faraday’s constant (96,485 C mol^−1^) and N_O_ the number of moles of the O compound.

Amperometric detection generally provides high sensitivity and the response remains linear over a wide range of concentrations. For these reasons, they are more widespread than coulometric detectors. Compared to coulometric detection, electrolysis consumes only about 10% of the analyte. The value of the measured current in an amperometric detector depends on the geometry of the electrode used (Table 1).

Regardless of the detector used, the number of electrons exchanged determines the measured current during the analysis. In case of detecting 5-HT, the number of electrons exchanges is 2 during the oxidation, as shown in Figure 5.

For electroactive compounds, charge can be correlated with the moles of the considered compound, extending to concentration. It is crucial to emphasize that the number of electrons exchanged in the electrochemical reaction is contingent on the applied potential. If the overvoltage is insufficient, then the electrochemical reaction remains incomplete.

ECD has some limitations: this detector is not universal and compounds must be electroactive to be detectable. In the same domain, GABA, a neurotransmitter involved in neurodegenerative diseases [49,50], can be analyzed by HPLC-ECD. However, GABA is not electroactive and requires chemical derivatization. In the Panrod et al. study, two derivatization pathway involving aldehydes moieties have been employed to be able to detect GABA [51]. Therefore, the development of a single analytical method for all NTs with an electrochemical detector appears to be challenging in light of the mentioned limitations.

The two most frequently employed working electrode materials are GC and boron-doped diamond (BDD). These materials showcase diverse physicochemical characteristics: the BDD electrode boasts the largest electrochemical window, facilitating the detection of numerous electroactive compounds. However, as indicated in the study by Zhang et al. in 2016, a high overpotential seems to reduce the peak area of the analytes (Figure 6). The sensitivity of electrode passivation is higher for a BDD electrode compared to a GC electrode [52]. Consequently, it seems that reproducibility is superior with a GC electrode.

The GC electrode exhibits a narrower electrochemical window compared to the BDD electrode. Consequently, the analysis range is restricted, and the oxidation of water impedes the detection of compounds with high oxidation potentials [54]. Analyzing compounds with potentials involving reduction is, to our knowledge, impossible due to the reduction of dissolved oxygen [55]. In Figure 7a, the detection current is depicted as a function of the potential applied for monoamines NTs with a GC working electrode. Ultimately, the choice between the two electrodes depends on the application when the compounds have relatively low oxidation potentials. Then, it is preferable to use a GC working electrode.

To enhance the sensitivity of NTs detection, two optimizations are considered: firstly, increasing the surface-to-volume ratio of the electrochemical cell, and secondly, using a modified electrode material to enhance its physicochemical characteristics. In the study by Xu et al., the working electrode on GC is modified through an electrochemical reaction with poly(para-aminobenzoic acid) (P-pABA) (Figure 7b) [56]. With a 0.7 V vs. Ag/AgCl potential detection, the sensitivity of measurement improved from 8.5 nA to 23 nA, 11 nA to 23 nA, 7 nA to 16 nA, 9 nA to 25 nA and 4 nA to 23 nA for 5-HIAA, 5-HT, DA, DOPAC and HVA, respectively. Augmenting sensitivity enhances the quantification of NTs.

In conclusion, the ECD is widely employed for the detection of monoamines due to its user-friendly nature and low maintenance cost. Its sensitivity enables the quantification of very low concentrations of NTs. To expand the capabilities of this detector, it would be possible to develop chemical derivatizations either directly on the compounds or on the electrode surface. These chemical modifications could induce a broader spectrum of applications.

### 3.4. Limit of Detection and Quantification

By convention, the limit of detection (LOD) and limit of quantification (LOQ) are established at signal-to-noise ratios of 3 and 10, respectively. The linear range (L.R.) denotes the measurement zone where the device-recorded signal is directly proportional to the quantity or concentration of the substance being measured.

Table 2 compiles the various values of LOD, LOQ, and the linear range for the NTs and their metabolites. Additionally, the different experimental conditions have been reported for informational purposes. The studies are ranked based on the particle size of the chromatographic C_18_ column used, from smallest to largest. The list of presented publications is not exhaustive. Other articles have been published on the use of HPLC-ECD but they did not detail the analytical development and LOD/LOQ of NTs. Those articles were not included in this summary table. For instance, we can cite several recent works by Guo et al. (2023) [57], Lei et al. (2023) [58], and Wang et al. (2022) [59].

According to these studies, the sensitivity of the targeted molecules can approach values close to 10^−9^ mol L^−1^ for HPLC system and less, close to 10^−11^ mol L^−1^, in UHPLC cases. As mentioned earlier, the use of columns with small particle sizes refines elution peaks and consequently lowers quantification limits. However, using such columns induces high back pressure. These low values facilitate the collection of microdialysis samples in the range of microliter, enabling real-time monitoring of neurotransmitter concentrations.

In electrochemical detection processes, the use of salts becomes crucial to increase conductivity and consequently achieve optimal performance. Typically, the aqueous phase of the mobile phase incorporates essential components, such as chelating compounds (e.g., sodium citrate, EDTA), which play a role in preventing coagulation [60], an anionic surfactant (sodium dodecyl sulfate, octane sulfonic acid) to prevent interference with lipids and enhance the retention of anionic compounds [61], and finally, a buffer (phosphate, citrate).

Most of the time, the pH of the mobile phase is buffered in an acidic environment. The more acidic the eluent, the faster the compounds will elute. Additionally, some compounds are sensitive and degrade more rapidly at basic pH. For instance, this is the case with DA, which oxidizes at a significant rate under basic pH conditions: the half-life at pH 5.6, 7.1, and 7.4 is 13.5 days, 19.7 min, and 4.95 min, respectively [62].

Moreover, the mobile phase, predominantly consisting of the aqueous phase with salts, is often augmented with methanol and/or acetonitrile. This strategic addition enhances the eluent’s apolar characteristics, ensuring a comprehensive and effective electrochemical detection setup.

**Table 2 molecules-29-00496-t002:** Recent values of separation, detection and quantification of compounds involved in serotonergic system and their application (non-exhaustive list). All values provided have been standardized to the same unit (mol L^−1^) for the purpose of comparability.

Reference		HPLC-ECD Conditions					Application	
Reinhoud et al. (2013) [30]	Methods and Application	C_18_ column (1.0 × 100 mm, 1.7 µm particles size) Flow rate = 50 µL min^−1^ Column temperature = 37 °C Injection volume = 5 µL Mobile phase: 100 mM phosphoric acid, 100 mM citric acid, 8 mM KCl, 0.1 mM EDTA and 2.8 mM OSA. ACN/H_2_O (8/92; *v*/*v*) pH = 3 Detection: GC working electrode (+0.65 V vs. Ag/AgCl)					Development of analytical method for the determination of monoamines concentrations and application to a microdialysis in a prefrontal cortex.	
	Detection (mol L^−1^)		5-HT	5-HIAA		DA	DOPAC	HVA
		LOD	8.3 × 10^−11^	3.5 × 10^−11^		4.2 × 10^−11^	5.0 × 10^−11^	4.7 × 10^−11^
Ferry et al. (2014) [63]	Methods and Application	C_18_ column (0.32 × 100 mm, 1.9 µm particles size) Flow rate = 8.5 µL min^−1^ Column temperature = 40 °C Injection volume = 1 µL Mobile phase: 140 mM potassium phosphate, 8 mM KCl, 0.1 mM EDTA, 6 mM OSA. MeOH/H_2_O (6/94; *v*/*v*) pH = 5 (adjusted with 10 mM NaOH) Detection: GC working electrode (+0.45 V vs. Ag/AgCl)					Development of analytical method for the determination of monoamines concentrations and application to microdialysis in the dorsal hippocampus	
	Detection (mol L^−1^)		5-HT			DA	3-MT	
		LOD	1.5 × 10^−9^			7.5 × 10^−10^	1.5 × 10^−9^	
		LOQ	5.0 × 10^−9^			2.5 × 10^−9^	5.0 × 10^−9^	
Schou-Pedersen et al. (2016) [64]	Methods and Application	C_18_ column (4.6 × 100 mm, 2.6 µm particles size) Flow rate = 0.8 mL min^−1^ Column temperature = 30 °C Injection volume = 20 µLMobile phase: 70 mM potassium dihydrogen phosphate, 2 mM OSA, 0.1 mM EDTA. MeOH/H_2_O (10/90; *v*/*v*) pH = 3.12 (adjusted with 1 M citric acid) Detection: Porous graphite working electrode (+0.40 V vs. Pd)					Development of chromatographic method for the quantification of monoamine NTs from sub-regions of guinea pig brain (intracellular and extracellular).	
	Detection (mol L^−1^)		5-HT	5-HIAA		DOPAC	DA	HVA
		LOQ	8.8 × 10^−9^	3.8 × 10^−9^		3.6 × 10^−9^	1.0 × 10^−8^	1.2 × 10^−8^
		L.R.	8.8 × 10^−9^–1.0 × 10^−6^	3.8 × 10^−9^–1.0 × 10^−6^		3.6 × 10^−9^–1.0 × 10^−6^	1.0 × 10^−8^–7.5 × 10^−7^	1.2 × 10^−8^–5.0 × 10^−7^
Van Dam et al. (2014) [65]	Methods and Application	C_18_ column (1.0 × 250 mm, 3.0 µm particles size) Flow rate = 40 µL min^−1^ Column temperature = from 30 °C to 36 °C Injection volume = 50 µL Mobile phase: 8 mM KCl, 50 mM phosphoric acid, 50 mM citric acid, 0.1 mM EDTA, from 1.8 to 2.2 mM OSA. MeOH/H_2_O (from 13/87 to 17/83; *v*/*v*) pH = from 3.0 to 3.6 (adjusted with NaOH) Detection: GC working electrode (from +0.63 V to +0.67 V vs. Ag/AgCl)					Development of analytical method for the quantification of biogenic amines and metabolites in human brain tissue.	
	Detection (mol L^−1^)		5-HT	5-HIAA		DOPAC	DA	HVA
		L.R.	5.3 × 10^−10^–1.8 × 10^−7^	2.2 × 10^−10^–7.7 × 10^−8^		2.8 × 10^−10^–1.6 × 10^−7^	2.7 × 10^−10^–1.8 × 10^−7^	5.8 × 10^−10^–3.2 × 10^−7^
Pantiya et al. (2024) [66]	Methods and Application	C_18_ column (3.0 × 500 mm, 2.6 µm particles size) Flow rate = 0.5 mL min^−1^ Column temperature = 25 °C Injection volume = 25 µLMobile phase: 130 mM sodium phosphate monobasic, 20 mM orthophosphoric acid, 2 mM sodium dodecyl sulfate, 50 µM EDTA. ACN, MeOH, H_2_O (5/10/95; *v*/*v*/*v*) pH = 3.2 (adjusted with phosphate buffer) Detection: GC working electrode (+0.50 V vs. Ag/AgCl)					Development of analytical method for the quantification of NTs and metabolites in brain mice microdialysates.	
	Detection (mol L^−1^)		5-HT	5-HIAA		DA	HVA	
		LOD	3.1 × 10^−10^	5.4 × 10^−10^		3.7 × 10^−10^	4.1 × 10^−10^	
		LOQ	3.9 × 10^−10^	5.8 × 10^−10^		4.2 × 10^−10^	4.3 × 10^−10^	
Allen et al. (2017) [67]	Methods and Application	C_18_ column (3.2 × 150 mm, 3.0 µm particles size) Flow rate = 0.6 mL min^−1^ Column temperature = N/A Injection volume = 40 µLMobile phase: 100 mM sodium acetate, 20 mM citric acid, 0.38 mM sodium octyl sulfate, 0.15 mM EDTA. ACN/H_2_O (5/95; *v*/*v*) pH = 3.3 (adjusted with glacial acetic acid) Detection: Dual working electrode (−0.22 V and +0.375 V)					Development of analytical method for the quantification of monoamines and metabolites in brain tissue of mice.	
	Detection (mol L^−1^)		5-HT	5-HIAA		DOPAC	DA	HVA
		LOD	1.7 × 10^−9^	6.5 × 10^−10^		7.4 × 10^−10^	8.2 × 10^−10^	4.1 × 10^−10^
		LOQ	3.6 × 10^−9^	1.3 × 10^−9^		1.5 × 10^−9^	1.6 × 10^−9^	1.4 × 10^−9^
		L.R.	3.6 × 10^−9^–1.7 × 10^−4^	1.3 × 10^−9^–7.9 × 10^−5^		1.5 × 10^−9^–8.9 × 10^−5^	1.6 × 10^−9^–2.0 × 10^−4^	1.4 × 10^−9^–8.2 × 10^−5^
Yardimci et al. (2023) [68]	Methods and Application	C_18_ column (4.6 × 250 mm, 5.0 µm particles size) Flow rate = 1 mL min^−1^ Column temperature = 36 °C Injection volume = 20 µL Mobile phase: 35 mM citric acid, 19 mM sodium citrate, 0.16 mM EDTA, 1.1 mM heptasulfonic acid. Glacial acetic acid/tetrahydrofuran/MeOH/H_2_O (0.11/0.3/2.5/97.085; *v*/*v*/*v*/*v*) pH = 4.9 (adjusted with 10 M NaOH) Detection: GC working electrode (+0.50 V vs. Ag/AgCl)					Analysis in hypothalamic and subcortical nuclei	
	Detection (mol L^−1^)		5-HT	5-HIAA		DA	DOPAC	
		LOQ	5.7 × 10^−7^	5.2 × 10^−7^		6.5 × 10^−7^	6.0 × 10^−7^	
Du et al. (2018) [69]	Methods and Application	C_18_ column (4.6 × 250 mm, 5.0 µm particles size) Flow rate = 1 mL min^−1^ Column temperature = 25 °C Injection volume = 20 µL Mobile phase: 25 mM sodium citrate, 0.01 mM EDTA. ACN/H_2_O (5/95; *v*/*v*) pH = 4.5 (adjusted with 1 M acetic acid) Detection: BDD working electrode (+0.70 V vs. Ag/AgCl)					Analytical method development	
	Detection (mol L^−1^)		5-HT				5-HIAA	
		LOD	2.1 × 10^−8^				1.6 × 10^−8^	
		LOQ	2.8 × 10^−8^				4.2 × 10^−8^	
		L.R.	2.8 × 10^−8^–1.1 × 10^−6^				2.6 × 10^−8^–2.6 × 10^−6^	
Zhang et al. (2016) [53]	Methods and Application	C_18_ column (4.6 × 250 mm, 5.0 µm particles size) Flow rate = 1 mL min^−1^ Column temperature = 25 °C Injection volume = 20 µL Mobile phase: 25 mM sodium citrate, 0.01 mM EDTA. ACN/H_2_O (5/95; *v*/*v*) pH = 4.5 (adjusted with 1 M acetic acid) Detection: BDD working electrode (+0.70 V vs. Ag/AgCl)					Determination of the concentrations of monoamines NTs in rat cortex and hippocampus tissues.	
	Detection (mol L^−1^)		5-HT	5-HIAA	DA	DOPAC	3-MT	HVA
		LOD	2.3 × 10^−8^	1.1 × 10^−8^	2.6 × 10^−8^	1.2 × 10^−8^	3.6 × 10^−8^	2.7 × 10^−8^
		LOQ	8.5 × 10^−8^	3.1 × 10^−8^	9.8 × 10^−8^	4.8 × 10^−8^	1.2 × 10^−7^	8.2 × 10^−8^
		L.R.	8.5 × 10^−8^–1.4 × 10^−6^	3.1 × 10^−8^–7.9 × 10^−7^	9.8 × 10^−8^–2.3 × 10^−6^	6.0 × 10^−8^–3.0 × 10^−6^	1.2 × 10^−7^–1.8 × 10^−6^	8.2 × 10^−8^–1.4 × 10^−6^
Jiang et al. (2015) [70]	Methods and Application	C_18_ column (4.6 × 250 mm, 5.0 µm particles size) Flow rate = 1 mL min^−1^ Column temperature = 30 °C Injection volume = 10 µL Mobile phase: 50 mM potassium dihydrogen phosphate, 0.1 mM octane sulfonic acid. MeOH/H_2_O (5/95; *v*/*v*) pH = N/A Detection: GC working electrode (+0.70 V vs. Ag/AgCl)					Development of the analytical method for monoamines NTs in human urine.	
	Detection (mol L^−1^)			5-HT			DA	
		LOD		6.1 × 10^−8^			3.9 × 10^−8^	
Lokhande et al. (2022) [71]	Methods and Application	C_18_ column (4.6 × 250 mm, 5.0 µm particles size) Flow rate = 1.3 mL min^−1^ Column temperature = 35 °C Injection volume = 20 µL Mobile phase: 50 mM potassium dihydrogen phosphate, 0.99 mM SOS and 53 µM EDTA. MeOH/H_2_O (12/88; *v*/*v*) pH = 2.5 (adjusted with 85% phosphoric acid) Detection: BDD working electrode (+0.70 V vs. Ag/AgCl)					Development of the analytical method for metabolites quantification in human CSF.	
	Detection (mol L^−1^)			5-HIAA			HVA	
		LOQ		6.5 × 10^−8^			6.9 × 10^−8^	
		L.R.		6.5 × 10^−8^–2.6 × 10^−6^			6.9 × 10^−8^–2.7 × 10^−6^	

Abreviations: N/A: not available, LOD: limit of detection, LOQ: limit of quantification, L.R.: linear range, EDTA: ethylendiaminetetraacetic acid, OSA: octane sulfonic acid, SOS: sodium octane sulfonate, ACN: acetonitrile, MeOH: methanol.

## 4. Conclusions

To study the brain serotonergic and dopaminergic systems, researchers employ intracerebral microdialysis coupled with high-performance liquid chromatography (HPLC) and electrochemical detector (ECD). HPLC, especially in reverse phase conditions, is a widely used separation technique for NT analysis, offering rapid and selective results. Electrochemical detection, particularly with glassy carbon electrodes, proves to be a sensitive and cost-effective method for quantifying NTs and their metabolites with values close to 10^−9^ mol L^−1^ for HPLC and 10^−11^ mol L^−1^ with UHPLC.

Optimizations, such as increasing the surface-to-volume ratio of the electrochemical cell and modifying electrode materials can enhance sensitivity. The determination of limits of detection and quantification, as well as the linear range, is crucial for evaluating the analytical performance of the HPLC-ECD system. Overall, the combination of microdialysis, HPLC, and ECD provides a powerful approach for understanding the dynamics of NTs and their metabolites, contributing to advancements in neuroscientific research and potential clinical applications.

It should be noted, however, that the use of ECD for neurotransmitters is becoming less common. Indeed, mass spectrometry is becoming more affordable, and the preventive maintenance of the analyzer is becoming simpler. Consequently, due to its high sensitivity and selectivity, mass spectrometry is currently the most common detector in analysis laboratories.

Research perspectives in the field of high-performance liquid chromatography with an electrochemical detector offer a fertile ground for improving the sensitivity and specificity of neurotransmitter detection. One promising avenue involves exploring the modification of the electrochemical detector’s surface. By altering surface properties, such as roughness or chemical composition, one can optimize the interaction with neurotransmitters, thereby enhancing detection efficiency.

Another research direction focuses on modifying the characteristics of the electrochemical cell itself, including its surface and dimensions. By adjusting these parameters, the signal-to-noise ratio can be increased, which is crucial for precise measurements. Electrochemical cells with an increased surface area can enhance the likelihood of interaction with analytes, thereby improving detection. Moreover, optimized dimensions can facilitate a clearer electrochemical response and reduce unwanted interferences.

By combining these approaches, it becomes conceivable to achieve quantification limits comparable to those obtained with mass spectrometry detectors, paving the way for significant advancements in understanding neurochemical mechanisms. These research endeavors hold the promise of bringing substantial improvements to neurotransmitter detection, opening new perspectives for understanding neurological disorders and mental illnesses.

## Figures and Tables

**Figure 1 molecules-29-00496-f001:**
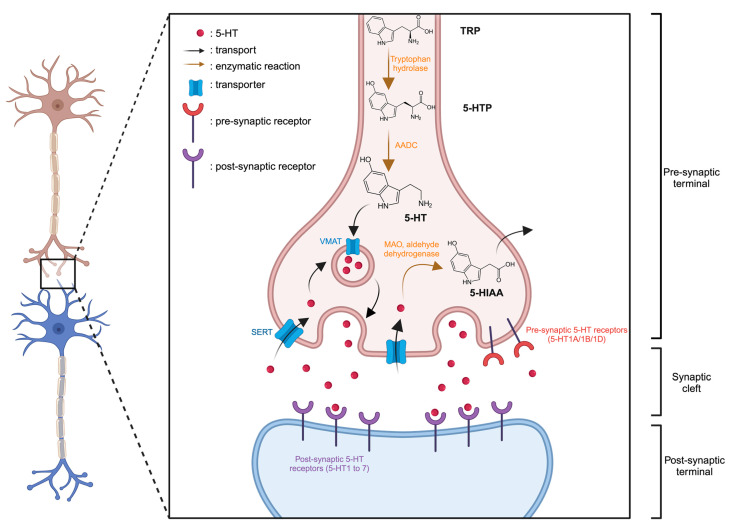
Representation of serotonergic synapse. Created with BioRender.com accessed on 15 January 2024.

**Figure 2 molecules-29-00496-f002:**
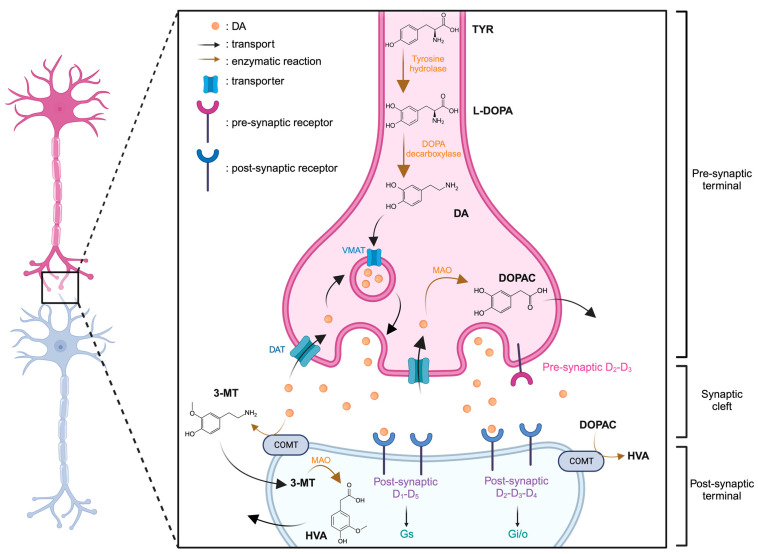
Representation of dopaminergic synapse. Created with BioRender.com accessed on 15 January 2024.

**Figure 3 molecules-29-00496-f003:**
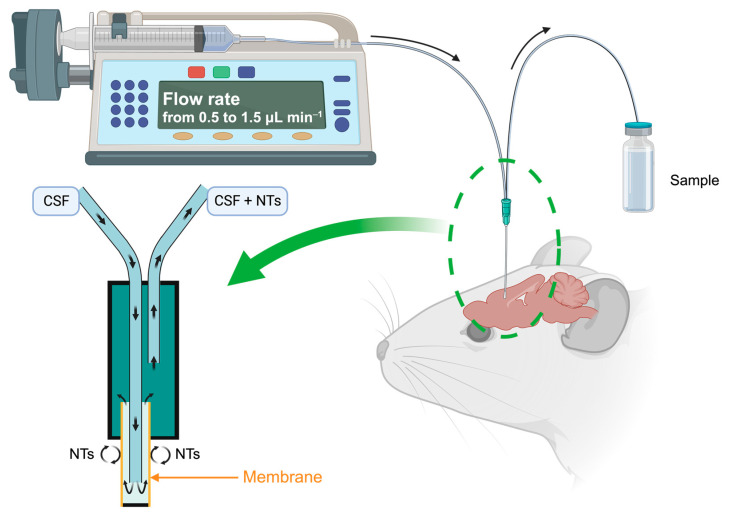
Schematic principle of the microdialysis on a brain mouse. Created with BioRender.com accessed on 15 January 2024.

**Figure 4 molecules-29-00496-f004:**
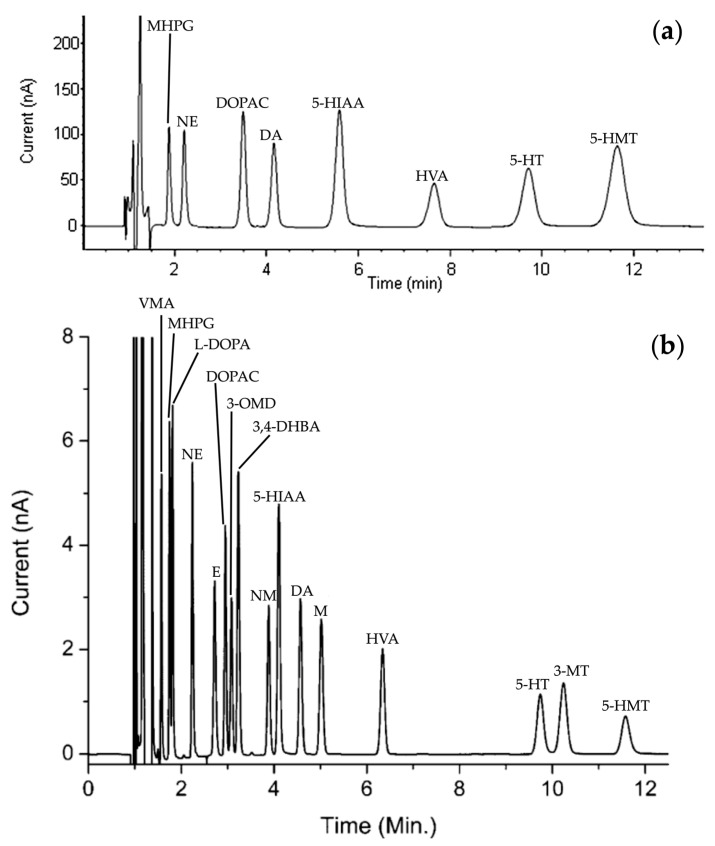
HPLC chromatogram reprinted and adapted with permission from [29] Copyright 2016 John Wiley and Sons (**a**) and UHPLC chromatogram reprinted and adapted with permission from [30] Copyright 2013 American Chemical Society (**b**) of a standards solution containing of each NTs and metabolites: DOPAC, DA, 5-HIAA, HVA, 3-MT and 5-HT. For information, other NTs and metabolites have been presented: 3-methoxy-4-hydroxyphenylglycol (MHPG), norepinephrine (NE), 5-hydroxymethyl tolterodine (5-HMT), D,L-4-hydroxy-3-methoxy-mandelic acid (VMA), L-3,4-dihydroxyphenylalanine (L-DOPA), epinephrine (E), 3,4-dihydroxybenzylamine hydrobromide (3,4-DHBA), 3-methoxy-L-tyrosine (3-OMD) and D,L-metanephirine (M).

**Figure 5 molecules-29-00496-f005:**
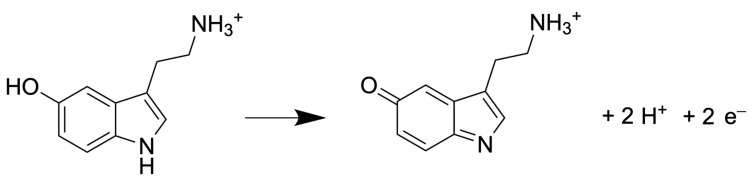
Half reaction of 5-HT oxidation [48].

**Figure 6 molecules-29-00496-f006:**
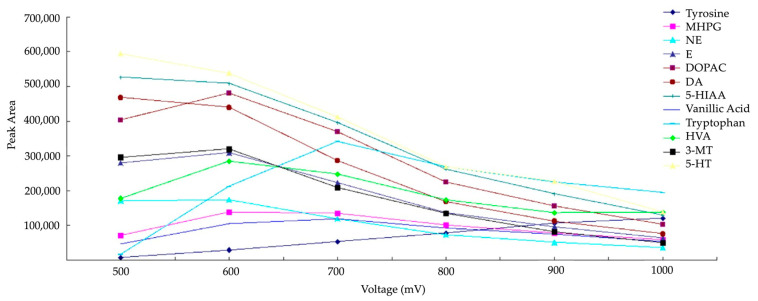
Hydrodynamic voltammograms of monoamines NTs on BDD working electrode reprinted and modified with permission from [53] Copyright 2016 Elsevier.

**Figure 7 molecules-29-00496-f007:**
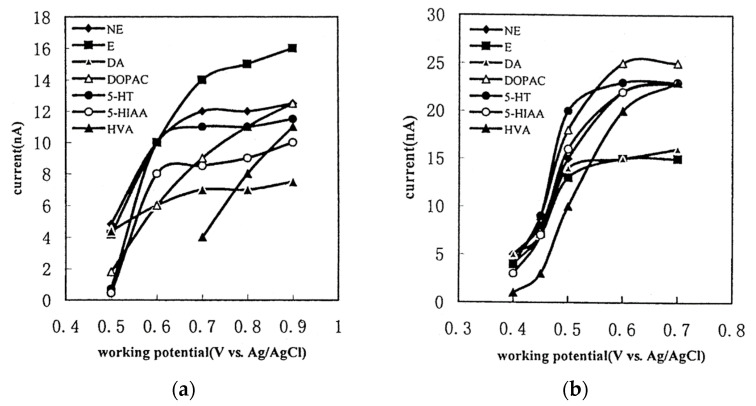
Hydrodynamic voltammograms of monoamines NTs and their main metabolites on GC working electrode (**a**) and P-pABA modified working electrode (**b**) reprinted with permission from [56] Copyright 2001 Elsevier.

**Table 1 molecules-29-00496-t001:** Limiting current equation as a function of geometry electrode used during the amperometric detection [47].

Electrode Geometry	Limiting Current Equation
Tubular	$i=1.61nFC_{O}{\left(\frac{DA}{r}\right)}^{2/3}\nu^{1/3}$
Planar, parallel flow in channel	$i=1.47nFC_{O}{\left(\frac{DA}{b}\right)}^{2/3}\nu^{1/3}$
Planar, perpendicular flow	$i=0.903nFC_{O}D^{2/3}\nu^{-1/6}A^{3/4}U^{1/2}$
Wall jet	$i=0.898nFC_{O}D^{2/3}\nu^{-5/12}a^{-1/2}A^{3/8}\nu^{3/4}$

With: i the intensity (A), C_O_ concentration of the O compound (mol cm^−3^), D the diffusion coefficient (cm^2^ s^−1^), A the electrode area (cm^2^), r radius of the tubular electrode (cm), ν average volume flow rate (cm^3^ s^−1^), b channel height (cm), U flow velocity (cm s^−1^) and a diameter of jet intel (cm).

## Data Availability

No new data were created or analyzed in this study. Data sharing is not applicable to this article.

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
