# Peer review of "The High-Precision Liquid Chromatography with Electrochemical Detection (HPLC-ECD) for Monoamines Neurotransmitters and Their Metabolites: A Review"

_molecules, 2024, doi:10.3390/molecules29020496_

Round 1

Reviewer 1 Report

Comments and Suggestions for Authors

This review described HPLC-ECD systems for determining monoamines neurotransmitters and their metabolites in biological samples.  Basically, it is well-written and can be accepted for publication of Molecules as the Special Issue.  However, before the publication, respond to the following comments.  

1)  Fig. 4(b):  There are many peaks which are identified constituents in this review.  Readers are interested in what these unidentified peaks are because this chromatogram obtained from a standard solution.  Because the constituents of these peaks were identified in the cited paper [29], the many peaks should be identified in Fig. 4(b).  Or, if you do not, you had better replace the Fig. 4(b) in your review to Fig. 1 in the cited paper [29] instead of Fig. 2 [29].     

2)  Page 6, lines 152-159:  You described the principle of quantitative analysis using coulometric detection in HPLC-ECD.  However, HPLC-ECD systems cited in Table 1 are used an amperometric detection, except for the reference of No. 53.  Add the principle of quantitative analysis using amperometric detection in HPLC-ECD. 

3)  Page 7, lines 188-190:  Because you describe “Analyzing compounds with potentials involving reduction is infrequent ---”, you should describe NTs which are detected by electrochemical reduction. 

4)  Page 8, lines 231-235:  Describe the reasons why chelating compounds and a surfactant are used in a mobile phase of HPLC-ECD for determining NTs.    

5)  Page 8, lines 231-235:  Describe the reasons why pH of mobile phase are maintained ranging from 2.5-5.0 in HPLC-ECD for determining NTs. 

6)  Table 1:  The unit of applied potential is corrected to “V vs. Ag/AgCl,” not “V/Ag/AgCl”.  Please reflect this correction throughout this review.  

Author Response

Dear reviewer, 

Please find in the attached file, our responses.

Best regards,

Reviewer 2 Report

Comments and Suggestions for Authors

Some improvements are needed prior to acceptance:

1.       In 3.1 section title " Gaz" should be "gas", please check and correct throughout the text; GABA, please provides full spellings for the first time in terms of specific terminology.

2.       On amino-ending neurotransmitters electrochemical detections, some reviews are referring to expand and clarify the background and discussion:  10.1016/j.ccr.2023.215487 ; 10.1016/j.mtadv.2022.100340;  

3.       Please specify how your LOD, L.R, LOQ were obtained by multiple measurement calculations (also demonstrate your experimental data).

4.       Please give some comparisons between this work and other standard ones when you were sampling and detecting.

5.       In general, the review draft could be expanded extensively on content and intensively on specific neurotransmitter. 

Comments on the Quality of English Language

Minor revision.

Author Response

Dear reviewer, 

Please find in the attached file, our responses to your comments.

Best regards,

Reviewer 3 Report

Comments and Suggestions for Authors

In this paper, the advantages of high performance liquid chromatography with an electrochemical detector in detecting and quantifying biological samples obtained by intracerebral microdialysis was reviewed. It mainly focused on monoamine neurotransmitters, sample acquisition techniques——microdialysis experiments, and HPLC-ECD technology. The main problems are as follows:

1. As a review article, its content is relatively limited, which is limiting the detection objects, sample acquisition techniques and detection techniques. Therefore, the availability of relevant literature is limited and the references cited in this paper are also outdated. It is recommended to start with the development of HPLC-ECD,  introduce the development of its application, as well as the new levels, trends and discoveries of this technology.

2. As is well known, HPLC is an excellent separation method, while ECD has the advantages of high efficiency and low cost. However, in the absence of standard samples, the combined technique has significant limitations in the qualitative and specific analysis of samples. It is suggested that the author focus on the detection of monoamine neurotransmitters, summarize the different detection technologies and methods that have developed with technological progress, compare the advantages and disadvantages of various detection technologies, and make the article hierarchical and developmental.

Comments on the Quality of English Language

Moderate editing of English language required

Author Response

(The authors gave the same response as above.)

Round 2

Reviewer 3 Report

Comments and Suggestions for Authors

As the author states, with the continuous development of instrument technology, the advantages of various large-scale equipment are gradually becoming more prominent and surpassing electrochemical methods, and publications on electrochemical detection are showing a declining trend. However, ECD still has outstanding advantages in terms of convenience and economy.

Based on the previously proposed suggestions, the author made appropriate modifications, compared the advantages and disadvantages of various methods, resulting in a more comprehensive review of the content.

Comments on the Quality of English Language

No comments.

Author Response

Dear reviewer,

Thank you for your comment, 

Best regards,